# Effects of antioxidants on cancer progression

Sarah Schmidt ⓘ , Xi Qiao ⓘ ✉ & Martin O Bergö ⓘ ✉

## Abstract

Antioxidant supplements are widely marketed for their promised health benefits, including cancer prevention and therapy support. This belief stems from the idea that excessive levels of reactive oxygen species (ROS) cause oxidative damage to cellular macromolecules. However, the effects of antioxidants are highly context-dependent and influenced by the compound type, dosage, and cancer type. While antioxidants may slow tumor progression in specific cancers such as MYC-driven lymphoma, they can accelerate tumor growth, metastasis, and angiogenesis in other settings, including KRAS-driven lung cancer and BRAF-driven melanoma. Some antioxidants may also influence the immune system in ways that support cancer therapies, such as immune checkpoint blockade. Here, we review recent studies that highlight the complex roles of antioxidants in cancer progression and discuss their potential implications for clinical practice.

## Introduction

Antioxidants are molecules that reduce intracellular ROS levels. A widely accepted definition is "any substance that delays, prevents, or removes oxidative damage to a target molecule" (Halliwell, 2024). ROS are produced during oxygen-dependent metabolic processes in mitochondria, the endoplasmic reticulum, and peroxisomes (Gorrini et al, 2013). They include both free radicals with unpaired electrons and non-radicals with paired electrons (Halliwell, 2024). At physiological levels, ROS regulate essential cellular signaling pathways, and play important roles in immune responses, for example, by enabling immune cells to eliminate pathogens (Halliwell, 2024).

Due to their high metabolic activity, cancer cells produce large amounts of ROS, which can lead to oxidative stress, damage to DNA, proteins, and lipids, and cell death (Gorrini et al, 2013). To maintain redox balance, both normal and malignant cells use various antioxidant systems (Halliwell, 2024). Endogenous antioxidants include enzymes, including catalases and peroxidases, and small molecules, including glutathione (GSH) and NADPH (Cheung and Vousden, 2022). These cell-intrinsic antioxidants are largely regulated by nuclear factor erythroid 2-related factor 2 (NRF2), a transcription factor which is often constitutively activated in cancer cells due to mutations in NRF2 itself or its negative regulator Kelch-like ECH-associating protein 1 (KEAP1) (Cheung and Vousden, 2022; Gorrini et al, 2013).

Exogenous antioxidants include pharmacological agents with redox-modulating abilities, such as the cysteine donor *N*-acetylcysteine (NAC), and dietary antioxidants, such as vitamins A, C, and E, and minerals (Halliwell, 2024). These are present in food, dietary supplements, and cosmetic products (Müller et al, 2022; Halliwell, 2024). Interestingly, some compounds, such as vitamin C, can switch roles and act as pro-oxidants at high concentrations (Ngo et al, 2019).

The idea that antioxidants can prevent or treat cancer is mainly based on the damaging potential of ROS and their role in cancer initiation through pro-mutagenic signaling and DNA damage (Gorrini et al, 2013). This idea has contributed to widespread antioxidant use among cancer patients and high-risk individuals. However, large-scale studies have produced conflicting and even disconcerting results. Some randomized clinical trials demonstrated increased cancer incidence among individuals taking antioxidants, including vitamin E and β-carotene (Klein et al, 2011; Omenn et al, 1996; Alpha-Tocopherol BCCPSG, 1994).

In this perspective article, we review studies that challenge the earlier simplistic view of antioxidants as universally protective. From recent mechanistic studies, including our own findings, we explore how antioxidants can either suppress or promote tumor progression depending on the molecular and cellular context; and we discuss the potential clinical implications. Examples of how antioxidants can influence different tumorigenic processes are presented in Fig. 1.

## Antioxidants can promote or inhibit tumor progression depending on the cancer context

Research over the past decade has provided insights into the paradoxical effects of antioxidants in cancer progression. Using genetically engineered mouse models and human and mouse cancer cell lines, several groups have shown that antioxidants can accelerate tumor progression under numerous conditions (Chio et al, 2016; DeNicola et al, 2011; Romero et al, 2017; Zou et al, 2021; Sayin et al, 2014).

In KRAS- and BRAF-driven lung cancer models, dietary supplementation with NAC and vitamin E reduced ROS levels and oxidative DNA damage in tumors (Sayin et al, 2014). This suppression of oxidative stress also inhibited p53 activation, which in turn accelerated tumor growth and reduced survival (Sayin et al, 2014). Notably, these effects were absent in tumors lacking functional p53, suggesting that antioxidants help tumor cells avoid ROS-induced activation of tumor-suppressive pathways (Sayin et al, 2014).

These findings are consistent with results from several large clinical trials. The Alpha-Tocopherol, Beta-Carotene Cancer Prevention Study (ATBC) was stopped early after male smokers who received antioxidant supplements showed significantly higher incidence of lung cancer compared to the

Department of Medicine Huddinge, Karolinska Institutet, Stockholm, Sweden. ✉E-mail: xi.qiao@ki.se; martin.bergo@ki.se
https://doi.org/10.1038/s44321-025-00269-5 | Published online: 11 July 2025

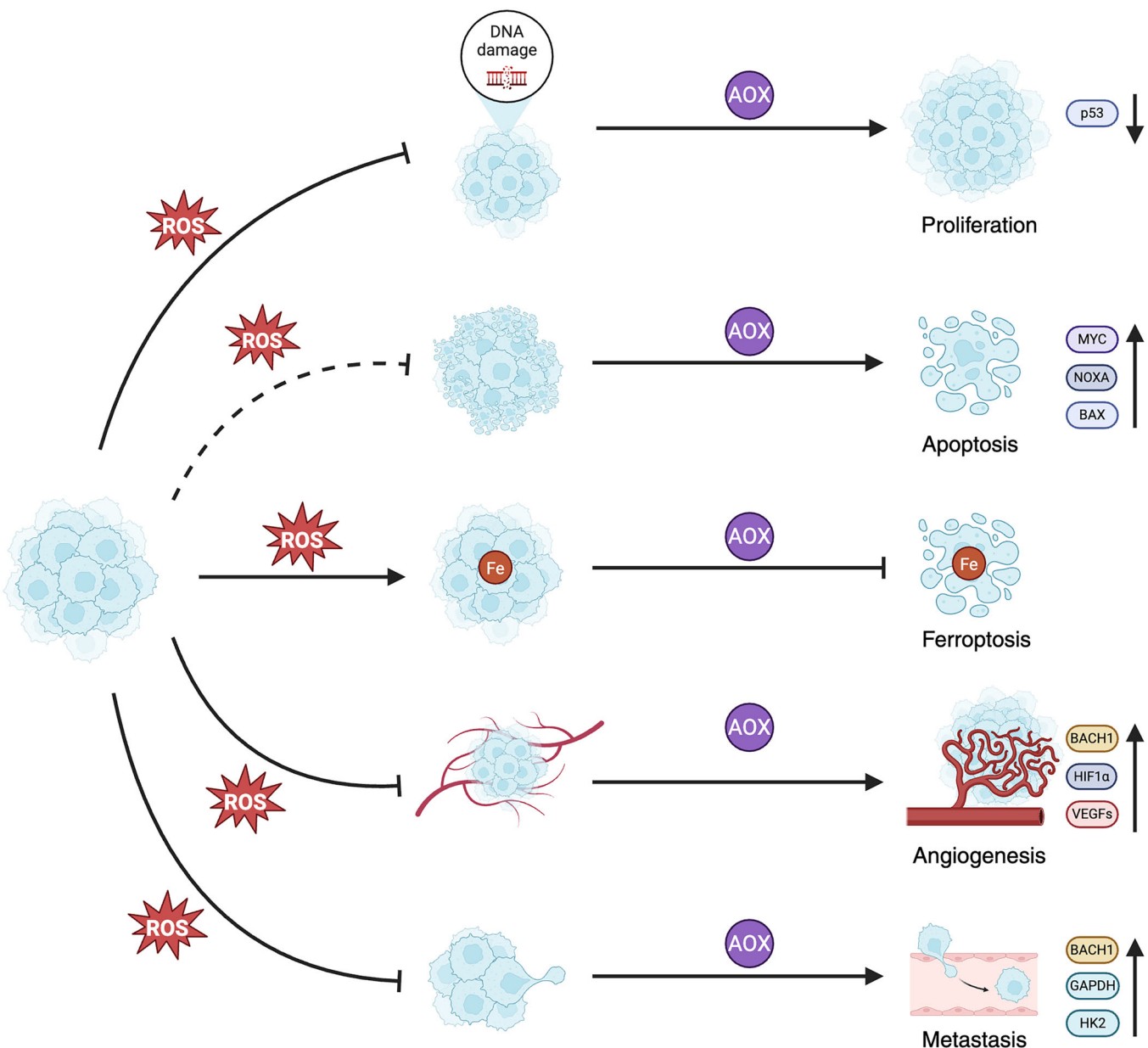

**Figure 1.    Pro- and anti-tumorigenic effects of antioxidants in cancer progression.**

Illustration of selected examples of how antioxidants—by lowering ROS levels—may either promote or block cancer progression. Antioxidants accelerate primary lung tumor growth by reducing levels of ROS-induced DNA damage and the tumor suppressor p53. ROS-lowering doses of antioxidants can also cause opposite effects and block the growth of certain tumors, such as B-cell lymphoma, through MYC-mediated activation of pro-apoptotic genes, including *NOXA* and *BAX*. During metastasis, antioxidants protect melanoma cells from ferroptosis—a form of cell death caused by iron (Fe)-dependent production of lipid-derived ROS. Antioxidants can also activate BACH1 and HIF1α—two transcription factors that increase the expression of glycolysis (e.g., *HK2* and *GAPDH*) and angiogenesis genes (e.g., *VEGFs*)—and thereby promote angiogenesis and metastasis in lung cancer. The figure was created with BioRender.com.

placebo group (Alpha-Tocopherol BCCPSG 1994; Duffield-Lillico and Begg, 2004). Similarly, the Beta-Carotene and Retinol Efficacy Trial (CARET) and the Selenium and Vitamin E Cancer Prevention Trial (SELECT) were stopped prematurely because participants receiving antioxidants had higher rates of lung and prostate cancer, respectively (Klein et al, 2011; Omenn et al, 1996). Based on our pre-clinical findings, we propose that these outcomes may reflect the accelerated progression of undiagnosed tumors—such as early lung lesions in smokers and prostate tumors in elderly men—driven by antioxidant supplementation.

A similar effect was observed in the *APC*^*Min/+* mouse model of familial adenomatous polyposis, a model for early-stage colorectal cancer (Zou et al, 2021). NAC and vitamin E increased the size of early tumors but had no effect on advanced tumors (Zou et al, 2021). However, histological analyses revealed that both early and

advanced tumors exhibited higher malignancy grades in antioxidant-administered mice, indicating that antioxidants promote tumor progression rather than initiation in this model (Zou et al, 2021).

In contrast, studies in MYC-driven B-cell lymphoma models demonstrated the opposite effect: ROS-lowering doses of antioxidants such as NAC and vitamin C reduced tumor growth, induced apoptosis, and increased survival (Yao et al, 2023). Mechanistic experiments linked this effect to the MYC-EGR1 transcription factor complex, and suggested that MYC over-expression represents a redox-sensitive vulnerability that can be therapeutically targeted with antioxidants in B-cell lymphoma (Yao et al, 2023). Similarly, earlier studies reported that NAC and vitamin C can reduce tumor growth in subcutaneous MYC-driven B-cell lymphoma (Gao et al, 2007). However, in those experiments, the tumor-suppressive antioxidant effect was linked to reduced hypoxia-inducible factor 1-alpha (HIF1α) levels, mediated by prolyl hydroxylase 2 (PHD2) and the von Hippel-Lindau (VHL) ubiquitin ligase (Gao et al, 2007). Moreover, in *Trp53*-knockout mice, which are prone to developing lymphoma, NAC reduced tumor incidence and increased survival—effects that were associated with reduced levels of ROS-induced DNA damage (Sablina et al, 2005).

These findings highlight the dual roles of antioxidants in cancer. Their impact depends on cancer type, stage, and genetic alterations. In tumors where redox-sensitive signaling promotes cell death, antioxidants can inhibit growth, while in others, antioxidants can accelerate growth by inhibiting stress-induced checkpoints.

Lastly, the site of ROS production may also influence antioxidant effects. Since mitochondria are the primary source of ROS in cancer cells (Gorrini et al, 2013), mitochondria-targeted antioxidants were tested in mouse models of BRAF-driven melanoma and KRAS-driven lung cancer (Le Gal et al, 2021). Administration of MitoQ and MitoTEMPO did not affect primary tumor growth or lymph node metastasis, nor did they influence levels of oxidative DNA damage (Le Gal et al, 2021). These results highlight the complexity of redox regulation in cancer and the limitations of using MitoQ and MitoTempo in these models.

## Antioxidants can promote metastasis

During metastasis, cancer cells must survive in hostile environments with high oxidative stress (Tasdogan et al, 2021). This oxidative barrier limits their chances of survival and reduces their ability to colonize distant organs (Tasdogan et al, 2021). However, both exogenous and endogenous antioxidants can reduce this stress and thereby increase the metastatic potential of cancer cells (Tasdogan et al, 2021).

Several studies have demonstrated that antioxidant supplementation increases metastasis in models of lung cancer and melanoma, without influencing primary tumor growth (Kashif et al, 2023; Le Gal et al, 2015; Lignitto et al, 2019; Piskounova et al, 2015; Wiel et al, 2019). In mice with melanoma, NAC increased lymph node metastasis, and in vitro, NAC and Trolox (a vitamin E analog) enhanced migration and invasion of human melanoma cells (Le Gal et al, 2015). Similarly, in KRAS-driven lung cancer models, both with and without p53, dietary antioxidants markedly increased lymph node and distant metastases (Wiel et al, 2019). Lung cancer cell lines derived from tumors of antioxidant-treated mice were also more invasive than control cell lines (Wiel et al, 2019).

A recent screen of redox-active compounds identified several dietary antioxidants, including vitamin C, β-carotene, retinyl palmitate, and canthaxanthin, which increased migration and invasion and lymph node metastasis of melanoma cells (Kashif et al, 2023). These results provide further evidence that ROS-lowering compounds can increase the ability of cancer cells to metastasize.

Mechanistic studies identified BTB and CNC homology 1 (BACH1), a redox-sensitive transcription factor, as a key mediator of the metastasis-promoting effects of antioxidants in lung cancer (Wiel et al, 2019). Antioxidants stabilize BACH1 which activates the expression of glycolysis-related genes, including *HK2* and *GAPDH*, and leads to increased glucose uptake, glycolysis rates, and lactate secretion; and ultimately, increased metastasis (Wiel et al, 2019). In melanoma, NAC-induced metastasis was associated with increased glutathione levels and activation of RHOA, a GTPase involved in cytoskeletal dynamics and cell migration and invasion (Le Gal et al, 2015).

Complementing these findings, recent studies highlighted the role of the tumor microenvironment in shaping metastatic success. Melanoma cells metastasize more efficiently through the lymphatic system than through the blood circulation, primarily because the blood exposes them to ferroptosis—an iron-dependent form of cell death (Ubellacker et al, 2020). Antioxidant administration protected circulating melanoma cells from ferroptosis, which enhanced their survival and metastasis through the blood (Ubellacker et al, 2020).

Other antioxidant compounds aside from NAC and vitamins have been implicated in metastasis. The anti-diabetic drugs saxagliptin and sitagliptin (DPP-4 inhibitors), as well as, α-lipoic acid, have demonstrated pro-metastatic abilities across multiple cancer types (Wang et al, 2016). These agents increased the expression of pro-metastatic proteins, including HIF1α and vascular endothelial growth factor (VEGF), and stabilized NRF2, which resulted in increased expression of endogenous antioxidants and reduced ROS levels in cancer cells (Wang et al, 2016). These results suggest that drugs not primarily classified as antioxidants can activate redox-regulating pathways and influence cancer progression.

## NAC and vitamins stimulate angiogenesis

Angiogenesis—the formation of new blood vessels—is essential for tumor growth and metastasis (Weis and Cheresh, 2011). Recent studies reveal that antioxidants such as NAC, vitamin C, and vitamin E can stimulate angiogenesis during KRAS-driven lung cancer progression (Wang et al, 2023). These antioxidants increase the expression of BACH1 and HIF1α and promote transcription of angiogenic genes (Wang et al, 2023). Notably, BACH1 was shown to regulate angiogenesis both in collaboration with and independently of HIF1α, highlighting its central role in antioxidant-induced tumor progression (Wang et al, 2023).

This mechanism was substantiated using multiple experimental models, including mouse lung tumor organoids, human lung cancer cell spheroids, and xenograft tumors (Wang et al, 2023). Antioxidant administration significantly upregulated angiogenesis-related genes in a BACH1-dependent fashion (Wang et al, 2023). Moreover, while

antioxidants increased HIF1α even under normoxia, hypoxia-induced BACH1 expression was found to occur independently of HIF1α (Wang et al, 2023). Indeed, prolyl hydroxylases were shown to regulate both HIF1α and BACH1, which links at least parts of their interplay to oxygen levels (Wang et al, 2023).

In vivo studies reinforced these findings. In immunodeficient mice, NAC and vitamin C stimulated angiogenesis in subcutaneous lung tumors, and this effect was abolished when BACH1 was knocked out in the cells (Wang et al, 2023). Moreover, subcutaneous lung tumors expressing high BACH1 levels were more sensitive to anti-angiogenic VEGFR2-targeting therapies (Wang et al, 2023). These findings establish BACH1 as a redox-sensitive transcription factor that orchestrates a pro-angiogenic program in response to antioxidants (Wang et al, 2023). They also suggest that antioxidant-induced tumor progression involves both increased glycolysis and angiogenesis (Wang et al, 2023; Wiel et al, 2019).

## Vitamins C and E: versatile molecules with potential therapeutic roles

Increasing evidence suggests that antioxidants may influence the efficacy of cancer therapies. Vitamin C, in particular, has been explored for its potential anticancer abilities in both experimental studies and clinical trials (Ngo et al, 2019). Some clinical trials also investigated vitamin E and β-carotene in combination with radiotherapy and chemotherapy, both of which are known to induce oxidative stress (Yasueda et al, 2016).

For example, a combination of vitamin E and β-carotene appeared to reduce side effects of radiotherapy in patients with head and neck cancer (Bairati et al, 2005). However, in smokers, vitamin E and β-carotene supplementation was associated with worse outcomes, including increased all-cause mortality and cancer recurrence (Meyer et al, 2008).

Early clinical studies investigating high-dose vitamin C in cancer patients reported conflicting results. Although some studies reported improved survival (Cameron and Pauling, 1976), others found no effect (Moertel Charles G. et al, 1985; Creagan et al, 1979). These discrepancies may be attributed to differences in administration routes, since intravenous vitamin C produces markedly higher plasma concentrations than oral intake (Padayatty et al, 2004).

Vitamin C is a molecule with multiple context-dependent functions (Ngo et al, 2019). For example, at micromolar concentrations, vitamin C acts as an antioxidant and neutralizes ROS, whereas at millimolar concentrations, it functions as a prooxidant and stimulates ROS production (Ngo et al, 2019). Moreover, vitamin C can modulate the activity of enzymes, such as iron-dependent and α-ketoglutarate-dependent dioxygenases (αKGDDs), by donating electrons (Ngo et al, 2019).

The prooxidant effect of vitamin C can be used therapeutically. In KRAS and BRAF-mutant colorectal cancer cells, vitamin C stimulated the uptake of its oxidized form, dehydroascorbate (DHA), through glucose transporters. This led to ROS accumulation, GSH depletion, and oxidative DNA damage, and subsequently reduced intestinal tumor development in mice (Yun et al, 2015). DHA-induced stress reduced glycolysis through S-glutathionylation and inactivation of GAPDH—a key glycolytic enzyme—leading to reduced ATP levels and AMPK activation. Antioxidants reversed vitamin C's anti-tumor effects (Yun et al, 2015), which highlights their interplay with redox regulation and metabolic activity.

Apart from its redox activity, vitamin C donates electrons to enzymes, such as ten-eleven-translocation (TET) methylcytosine dioxygenases, which catalyze DNA demethylation reactions (Ngo et al, 2019). In leukemia, TET2 loss-of-function mutations are associated with altered DNA methylation patterns and poor outcomes (Cimmino et al, 2017). Vitamin C administration restored TET2 activity, promoted DNA demethylation, and slowed leukemia progression in experiments with leukemia cell and mouse models (Cimmino et al, 2017). Notably, these effects were independent of ROS, suggesting that the epigenetic changes in leukemia cells are unrelated to vitamin C's anti- and prooxidant functions (Cimmino et al, 2017). Vitamin C administration also increased the expression of genes that facilitate DNA demethylation and repair processes, such as PARP, in leukemia cells (Cimmino et al, 2017). PARP inhibitors, which are commonly used to treat leukemia, demonstrated greater efficacy in eliminating leukemia cells when combined with vitamin C by causing DNA damage and cell cycle delays, and inducing

cell differentiation (Brabson et al, 2023; Cimmino et al, 2017).

Vitamin C has also shown promise as an adjuvant to immune checkpoint therapy (Magrì et al, 2020). In syngeneic breast, pancreatic, and colorectal mouse tumor models, high-dose vitamin C potentiated the effects of anti-PD-1 and anti-CTLA-4 antibodies, which was mediated by the presence of functional T cells in the tumor microenvironment (Magrì et al, 2020). Notably, co-administration with NAC reduced vitamin C-induced DNA damage without compromising anti-tumor efficacy, suggesting that vitamin C's effect on immune cells was independent of its redox activity (Magrì et al, 2020).

Recently, vitamin C was found to act as a protein modifier in human and mouse cancer cells (He et al, 2025). Vitamin C selectively bound lysine residues and prevented dephosphorylation of STAT1—an immune-regulatory transcription factor—which enhanced its nuclear localization and activated CD8+ T-cell-mediated anti-tumor immune responses (He et al, 2025). In a breast cancer mouse model, this mechanism enhanced the efficacy of PD-1 blockade, but only in immunocompetent mice (He et al, 2025).

Similarly, vitamin E in combination with immune checkpoint inhibitor therapy was associated with increased survival in melanoma patients and a heterogenous cohort of patients with colon, breast, or kidney cancer (Yuan et al, 2022). Moreover, in several syngeneic mouse tumor models, vitamin E enhanced the efficacy of immune checkpoint inhibitors alone or combined with cancer vaccines and chemotherapy (Yuan et al, 2022). Mechanistically, it was discovered that vitamin E can bind to the protein tyrosine phosphatase and cell-intrinsic immune checkpoint SHP1 in dendritic cells and prevent its activation (Yuan et al, 2022). SHP1 suppression activated dendritic cells and subsequently, CD8+ T-cell-mediated killing of cancer cells (Yuan et al, 2022). Less immunogenic tumors did not respond to vitamin E, indicating that vitamin E's ability to activate the anti-tumor T-cell response depended on dendritic cells (Yuan et al, 2022).

Together, these results suggest that vitamins C and E can potentiate anticancer therapies, often through mechanisms that are independent of their classical ROS-reducing properties. Their context-specific

roles highlight the need for precision in their potential therapeutic use.

## Conclusions

These studies highlight the complex and context-dependent effects of antioxidants on cancer. The effects of antioxidant compounds vary widely depending on factors such as compound type, dosage, route of administration, cancer type, disease stage, and genetic background. While antioxidants like NAC, vitamin C, and vitamin E can substantially promote tumor growth and metastasis in several contexts—particularly through BACH1 activation and stimulation of glycolysis and angiogenesis—they also show therapeutic potential in other contexts, including leukemia, MYC-driven B-cell lymphoma, and immunotherapy-responsive tumors.

Antioxidants affect cancer biology via several different mechanisms, such as modulation of redox homeostasis, alteration of metabolic pathways, regulation of transcription factors, epigenetic reprogramming, and immune response modulation. This functional heterogeneity demands clear-cut, evidence-based guidelines for antioxidant use, particularly in patients with cancer or increased cancer risk—where dietary supplementation seems to be particularly troublesome.

Future research should focus on understanding molecular mechanisms that underlie antioxidant effects in different cancer types and therapeutic contexts. Expanding these studies to hematologic malignancies and further studying antioxidant interactions within the tumor microenvironment may produce valuable insight. Moreover, clinical trials which account for cancer-specific and patient-specific parameters will be essential for informing therapeutic recommendations.

Given the well-documented risks associated with non-prescribed antioxidant supplementation, caution is clearly warranted. Until definitive data are available—and policy makers feel confident in making clear recommendations—cancer patients and individuals at risk should avoid antioxidant use outside of medically supervised protocols. Continued research will undoubtedly increase our understanding of antioxidants and their effects and ultimately allow us to more strategically harness antioxidant mechanisms.

## Peer review information

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

## Acknowledgements

The authors acknowledge financial support for these studies from the Swedish Cancer Society (No. 23 3146 Pj 01 H to MOB) and the Swedish Research Council (No. 2024-03027 to MOB).

## Author contributions

**Sarah Schmidt**: Conceptualization; Investigation; Writing—original draft. **Xi Qiao**: Conceptualization; Visualization; Writing—review and editing. **Martin O Bergö**: Conceptualization; Supervision; Funding acquisition; Writing—review and editing.

## Disclosure and competing interests statement

The authors declare no competing interests.

