## [Peer Review File · EMBO Molecular Medicine]

Effects of antioxidants on cancer progression

Sarah Schmidt, Xi Qiao, and Martin Bergö

Corresponding authors: Xi Qiao (xi.qiao@ki.se) , Martin Bergö (martin.bergo@ki.se)

Review Timeline:

Submission Date:	6th Feb 25
Editorial Decision:	28th Feb 25
Revision Received:	5th Jun 25
Accepted:	25th Jun 25

Editor: Lise Roth

Transaction Report:

28th Feb 2025

Dear Dr. Qiao,

Thank you for the submission of your Perspective to EMBO Molecular Medicine. We have now received feedback from the experts who agreed to evaluate your manuscript.

As you will see from the reports below, they overall found the article interesting and well written. They nevertheless make several suggestions to improve the interest and impact of your work.

We would therefore welcome a revised version of your manuscript that would address these points. Please note that our new Perspective format is flexible. As an indication, we recommend no more than 4 figures, 5000 words and 50 references.

Please attach a covering letter giving details of the way in which you have handled each of the points raised by the referees.

Additionally, please address the following editorial queries:

- Please clarify who is the main corresponding author, and update X. Qiao's account.
- Please note that all corresponding authors must have an ORCID identifier. Currently, an ORCID ID is missing for X. Qiao.
- Please enter the funding information in the submission system.
- Please make sure that the figure is referenced in the text.
- For the figure, please note:
 1. If there are certain aspects of your figure draft that are based upon assumptions or where the scientific data remains ambiguous, please add a comment so that we can work with you on an accurate depiction. Please ensure the directionality and nature of interactions is presented accurately.
 2. If the figure or part of the figure has been adapted from a published figure, please add this information to the figure legend (e.g., 'Adapted from...' or 'Based on...').
 3. If you use an image data base for scientific iconography (e.g., BioRender), please let us know if you have a license that allows for publication in an academic journal.

Looking forward to receiving your revised manuscript,

With kind regards,

Lise Roth

**** Reviewer's comments ****

Referee #1 (Bridging gap comments for Author):

The review does not clearly define the gap between bench and bedside and do not put forward interesting and feasible suggestions to narrow it. It does review a limited part of the literature on the topic.

Referee #1 (Remarks for Author):

This is a rather simple review summarizing the known effect of antioxidants on cancer progression. It's a well-written and fluent piece on the complex role of antioxidants in cancer. However, there are a few areas that could be improved.

Structure: The text is quite simple and superficial. Extending each section could improve interest for a wiser audience. Different types of cancer should be discussed.

While the text does mention both the positive and negative effects of antioxidants, it seems to lean more heavily on the negative side. Ensuring a balanced presentation of evidence would strengthen the article.

Figure: The only figure provided is rather simple and not properly embedded in the text. I encourage the authors to enrich the illustrations of this review.

There's some repetition of information, particularly about the effects of antioxidants on different cancer types. This could be better explained by adding more paragraphs.

Some sections could benefit from better transitions to improve flow between ideas.

The conclusion is good, but it could be strengthened by explicitly tying together the various threads discussed throughout the text.

Some technical terms might benefit from brief explanations for a broader audience.

Future directions: The conclusion touches on this, but a more detailed discussion of future and unforeseen research directions could enhance the article's impact.

Referee #2 (Bridging gap comments for Author):

Yes

Referee #2 (Remarks for Author):

This is a short review focused on the author's work in assessing the effects of antioxidants and BACH1 in cancer progression. The paper is clear and well written, and very nicely summarizes a series of papers published by this lab over the past few years. While there are studies showing that ROS can promote metastasis, the authors here focus on their own work demonstrating how antioxidants can function to regulate BACH1 and support metastasis.

The consequences of ROS regulation are broad and a focused review such as this one is very valuable. However, I think it would be useful for the authors to make this focus clear. The title could be made more specific and it's not clear why the Figure shows only some of the potential responses to antioxidants (for example, ferroptosis, the role of ROS promoting proliferation and metastasis, and the role of ROS in regulating the tumor microenvironment are not mentioned). The focus is fine but should be clarified.

To provide a little more balance, it might be helpful to cite the original work from other groups. As it stands, most of the primary research papers cited emanate from this group.

Point-by-point response to comments from Reviewer 1

We thank the Reviewer for their time, valuable comments and constructive criticism, which have helped us to improve the quality of our manuscript. We have addressed the comments, in most cases, through revised or new content. Newly added and substantially revised text is underlined in the revised manuscript for clarity.

Below is a point-by-point response (in normal font) to the Reviewer's comments (in bold font).

Reviewer #1 wrote that “The review does not clearly define the gap between bench and bedside and do not put forward interesting and feasible suggestions to narrow it. It does review a limited part of the literature on the topic.”

We appreciate your comment. In this Perspective, we highlight that the effects of antioxidants in cancer are multifaceted and context-dependent. The gap between bench and bedside stems to a great extent from inconsistent and often adverse results across experimental studies and randomized clinical trials. We have revised the conclusions section of this manuscript where we have tried to better emphasize the steps necessary to narrow this gap.

We agree that this work reviews only a fraction of the literature on the topic. Since we initially submitted it as a Commentary with limited references, we were not able to include more literature. However, we have changed the revised manuscript to a Perspective format, which has allowed us to extend the literature. To extend the scope of the review, we have now added a new section on recent discoveries regarding the diverse molecular functions of vitamins C and E and their effects in combination with certain cancer therapies (pages 7-10).

Structure: The text is quite simple and superficial. Extending each section could improve interest for a wiser audience. Different types of cancer should be discussed.

Thank you for the feedback. Since we initially submitted the manuscript as Commentary and now as Perspective, we aimed to present the content by using a more accessible and less technical language for a broader scientific audience.

The initial version of the manuscript covers antioxidant effects in lung cancer, prostate cancer, melanoma, colorectal cancer and lymphoma (pages 3-7). In the revised manuscript, we have added studies including other cancer types, such as head and neck cancer, leukemia, pancreatic cancer and breast cancer (pages 7-10).

While the text does mention both the positive and negative effects of antioxidants, it seems to lean more heavily on the negative side. Ensuring a balanced presentation of evidence would strengthen the article.

Thank you for this comment. In the revised manuscript, we have added more studies in which antioxidants counteract cancer progression (page 4). We have also added an entirely new section discussing studies that report positive effects of vitamins C and E to provide a more balanced and comprehensive perspective on the topic (pages 7-10).

Figure: The only figure provided is rather simple and not properly embedded in the text. I encourage the authors to enrich the illustrations of this review.

We failed to understand whether your meaning was that the figure was not properly embedded—i.e., pasted in—within the text or whether you mean that the figure was not called out properly in the text. However, we have revised both the figure and the figure legend (page 17) to better connect it with the corresponding paragraphs on pages 2-7. We

have also called out the figure in the final paragraph of the Introduction section (page 3). We hope that you agree to these changes.

There's some repetition of information, particularly about the effects of antioxidants on different cancer types. This could be better explained by adding more paragraphs.

In response to this comment, we have divided up some of the paragraphs (see pages 2,4-5 and 10) and also added numerous additional paragraphs on this topic (see pages 7-10).

Some sections could benefit from better transitions to improve flow between ideas.

Thank you for this suggestion. We have now addressed this important issue by revising the text, divided up some of the paragraphs, and connected them more logically. We hope that you agree (see pages 2-5 and 10).

The conclusion is good, but it could be strengthened by explicitly tying together the various threads discussed throughout the text.

Thank you for the comment. We have made substantial revisions to the original conclusions section and added a new paragraph to better connect the different aspects discussed throughout the manuscript.

Some technical terms might benefit from brief explanations for a broader audience.

We have briefly explained the majority of technical terms to facilitate easier reading for a broader audience. Please let us know if we have missed any.

Future directions: The conclusion touches on this, but a more detailed discussion of future and unforeseen research directions could enhance the article's impact.

We have now added new ideas to the discussion on future research in the conclusions section and hope you find it valuable (pages 10-11).

Thank you again for the thorough review. We hope the revised manuscript reflects the requested improvements and is now acceptable for publication.

Point-by-point response to comments from Reviewer 2

We thank the Reviewer for their time, thoughtful comments and constructive criticism, which have helped us to improve the quality of our manuscript. We have responded to the comments, in most cases, through revised or new content. Newly added and substantially revised text is underlined in the revised manuscript for clarity.

Below is a point-by-point response (in normal font) to your comments (in bold font).

Reviewer #2 wrote that “This is a short review focused on the author's work in assessing the effects of antioxidants and BACH1 in cancer progression. The paper is clear and well written, and very nicely summarizes a series of papers published by this lab over the past few years. While there are studies showing that ROS can promote metastasis, the authors here focus on their own work demonstrating how antioxidants can function to regulate BACH1 and support metastasis.”

Thank you for the positive feedback on our manuscript.

The title could be made more specific and it's not clear why the Figure shows only some of the potential responses to antioxidants (for example, ferroptosis, the role of ROS promoting proliferation and metastasis, and the role of ROS in regulating the tumor microenvironment are not mentioned). The focus is fine but should be clarified.

Thank you for your comments. The manuscript was initially submitted as Commentary; however, we have changed the format of the revised manuscript to a Perspective. We have now added more references reporting anti-tumorigenic roles of antioxidants (page 4) and an entirely new section presenting other functions of vitamins C and E and their potential roles in therapeutic settings (pages 7-10). Therefore, we have decided to keep the original, broader title, as the manuscript covers both pro- and anti-tumorigenic effects of antioxidants in cancer.

Regarding the figure, our intention was to illustrate some examples of tumor responses to antioxidants under high ROS conditions, as discussed in the manuscript. We tried to cover both tumor cell-intrinsic effects and effects on the tumor microenvironment, such as angiogenesis. In response to your comment, we have revised the figure and included antioxidant effects on ferroptosis. While we agree that ROS-driven tumor proliferation and metastasis are important, our idea was to focus primarily on antioxidant-mediated effects.

To provide a little more balance, it might be helpful to cite the original work from other groups. As it stands, most of the primary research papers cited emanate from this group.

Thank you for the feedback. In response, we have replaced some review articles on clinical trials with primary literature (page 2-3) and added studies from other groups to the sections on cancer progression (pages 3-4) and metastasis (page 6). We have also expanded the manuscript by adding a new section covering findings on vitamins C and E, all of which are contributions from other groups (pages 7-10).

Thank you again for the valuable comments on our manuscript. We hope you will find the revised version of the manuscript acceptable for publication.

25th Jun 2025

Dear Dr. Qiao,

Thank you for submitting your revised Perspective, which has been reviewed by referee #2. As you will see below, this referee is satisfied with the revisions, and I am pleased to inform you that your manuscript is now accepted for publication and is being sent to our publisher to be included in the next available issue of EMBO Molecular Medicine.

Your manuscript will be processed for publication by EMBO Press. It will be copy edited and you will receive page proofs prior to publication. Please note that you will be contacted by Springer Nature Author Services to complete licensing information.

This Perspective is free of charge, and we will shortly send you an email with a token. When you are contacted in a few weeks to sign your license agreement and review article proofs, please enter this token into the relevant field in the Springer Nature author services system.

With kind regards,

Lise Roth

Referee #2 (Remarks for Author):

The authors have improved the review in light of the comments.